# A Survey on the Development of Network Protocol Fuzzing Techniques

**Zhaowei Zhang [1], Hongzheng Zhang [1], Jinjing Zhao [1,\*] and Yanfei Yin [2]**

[1] National Key Laboratory of Science and Technology on Information System Security, Beijing 100101, China; xavierzh0710@gmail.com (Z.Z.); baiweidou31812631@gmail.com (H.Z.)

[2] China Aerospace Academy of Systems Science and Engineering, Beijing 100037, China; christmas720@163.com

\* Correspondence: zhaojinjing@nudt.edu.cn

**Abstract:** Network protocols, as the communication rules among computer network devices, are the foundation for the normal operation of networks. However, security issues arising from design flaws and implementation vulnerabilities in network protocols pose significant risks to network operations and security. Network protocol fuzzing is an effective technique for discovering and mitigating security flaws in network protocols. It offers unparalleled advantages compared to other security analysis techniques thanks to the minimal requirement for prior knowledge of the target and low deployment complexity. Nevertheless, the randomness in test case generation, uncontrollable test coverage, and unstable testing efficiency introduce challenges in ensuring the controllability of the testing process and results. In order to comprehensively survey the development of network protocol fuzzing techniques and analyze their advantages and existing issues, in this paper, we categorized and summarized the protocol fuzzing and its related techniques based on the generation methods of test cases and testing conditions. Specifically, we overviewed the development trajectory and patterns of these techniques over the past two decades according to chronological order. Based on this analysis, we further predict the future directions of fuzzing techniques.

**Keywords:** vulnerability discovery; network protocol; fuzzing; network security; network protocol security

## 1. Introduction

Network protocols are the foundation of computer networks. They define the format, meaning, order, and actions of message exchange among communication entities. With the development of network applications, vulnerabilities in network protocols have emerged as a critical factor threatening the security of networks.

In 2001, the "Code Red" worm exploited vulnerabilities in the HTTP protocol implementation, gaining superuser privileges on Microsoft IIS web servers. It infected approximately 360,000 servers and 1 million computers worldwide, and resulted in an estimated global loss of around USD 2.6 billion. In 2014, the "Heartbleed" vulnerability in OpenSSL was publicly disclosed and exploited [1]. This incident affected around 500,000 Internet servers. In 2021, a set of vulnerabilities named "WRECK" were disclosed [2], related to the implementation of DNS protocols, which could lead to denial of service or remote code execution. Over 180,000 devices in the United States alone were affected.

Regarding the discovery of system vulnerabilities, the concept of fuzzing was proposed by Professor Barton Miller at the University of Wisconsin in 1988 [3]. It has gradually evolved into an effective, fast, and practical technique [4–7]. The main idea is to develop a fuzzing tool, known as a fuzzer [8], capable of generating semi-valid data (test cases) and submitting them to the system under test (SUT) to find if any security issues exist. Semi-valid data refer to data that can be correctly received and processed by the target

system, uncovering deep-seated vulnerabilities that are difficult to detect through traditional means [9–12]. The workflow of fuzzing is illustrated in Figure 1, where the initial information is also referred to as the seed.

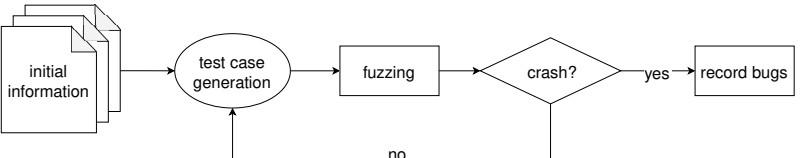

**Figure 1.** General workflow of network protocol fuzzing techniques.

As a widely adopted vulnerability detection technique, fuzzing has found extensive application in the field of protocol security testing. Currently, protocol fuzzing techniques primarily focus on testing protocol implementations. For example, in the CVE-2019-16519 vulnerability, during the BGP protocol communication process, sending a close communication with a sufficiently long message length can cause a buffer overflow error in the route daemon. In protocol fuzzing, if a fuzzer generates and sends a close communication packet that meets the length criteria, it may trigger this vulnerability. One key difference from traditional fuzzing techniques is that many protocols are stateful, requiring the implementation to receive a series of message requests and send appropriate responses based on the current state. In contrast, traditional fuzzing tools do not consider the software's state information or the structure and order of messages to be sent.

In the past two to three years, there have been no systematic reviews specifically focused on network protocol fuzzing techniques. If we go back a few more years, there are existing surveys on protocol fuzzing and traditional fuzzing that mention the application of fuzzing techniques to network protocols. Liang et al. [7] addressed the key challenges faced by traditional fuzzing techniques at each phase and provided an overview of the research conducted to address these challenges. The paper discusses the different application scenarios for fuzzing techniques and briefly describes two network protocol fuzzers. Li et al. [13] specifically emphasized coverage-based traditional fuzzing techniques and discussed technologies integrated in fuzzing. They also mentioned network protocol fuzzing techniques within different application scenarios, highlighting only four different fuzzers. Manes et al. [14] dissected the overall fuzzing process into several phases and explained the design choices of each phase using relevant techniques. In the input generation phase, they mentioned that some protocol fuzzing techniques adopt predefined models and inference-based methods for test case generation, without providing specific introductions for these fuzzers. Munea et al. [15] classified and compared protocol fuzzing techniques from five different perspectives, but the survey only includes five specific fuzzers. Hu and Pan [16] provided a summary of protocol fuzzing techniques in chronological order and introduced machine learning techniques applied to network protocol fuzzing. However, their study has limitations in terms of the comprehensiveness of the collected relevant techniques and the analysis conducted.

This paper aims to fill the existing gap by conducting a survey in protocol fuzzing area. Taking a chronological approach, we categorized and summarized approximately fifty protocol fuzzing and related research techniques based on the generation methods of test cases and testing conditions. This enables a clear overview of the development trajectory and patterns at different stages since the inception of the technique. Based on the analysis, we further predict the future directions of protocol fuzzing techniques.

The rest of this paper is structured as follows. In Section 2, we will outline our literature search methodology. In Section 3, we will introduce the fundamental knowledge and common classification methods related to network protocol fuzzing techniques. In Section 4, we will comprehensively review the development and advancements in network protocol fuzzing techniques over the past two decades, dividing them into different stages. Section 5 will examine and analyze the techniques related to network protocol fuzzing techniques and assess their contributions to the field. In Section 6, we will address key

issues in protocol fuzzing and evaluate the advantages and disadvantages of the current state-of-the-art techniques. In Section 7, we will provide a comprehensive analysis of the existing bottlenecks in protocol fuzzing techniques and offer insights into the future trends of the technique. Finally, we will conclude this in the Section 8.

## 2. Review Method

In order to conduct a comprehensive survey on network protocol fuzzing, in the following sections, we will introduce our research methods and collected data in detail.

### 2.1. Research Questions

This survey mainly aims to answer the following research questions about network protocol fuzzing.

1. RQ1: What are the key problems and the corresponding techniques in protocol fuzzing research?
2. RQ2: What are the state-of-the-art techniques and their pros and cons?
3. RQ3: What are the future directions of protocol fuzzing and related techniques?

RQ1, which is answered in Section 6, allows us to explore an in-depth view on protocol fuzzing. RQ2, which is answered in Section 6, is proposed to give an insight into the comparisons and suitable scenarios of existing techniques. Finally, based on the answer to the previous questions, we expect to identify the unresolved problems and future opportunities of protocol fuzzing and related techniques in response to RQ3, which is answered in Section 7.

### 2.2. Search Strategy

In order to provide a complete survey covering as many related papers as possible, we conducted a search for relevant techniques through three steps. First, we searched some main online repositories such as IEEE XPlore, ACM Digital Library, USENIX, Springer Online Library, etc., and conducted a literature search to collect papers that utilize the terms "fuzz testing", "fuzzing", or "fuzzer" in conjunction with "protocol", as well as papers that include "Protocol state machine", "FSM", or "Protocol Modeling" in their titles, abstracts, or keywords. Second, we used abstracts of the collected papers to exclude some of them based on the following selection criteria:

1. Not related to the network protocol field;
2. Not written in English;
3. Not accessible via the Web.

Third, we verified the references of the collected paper to determine if there were any overlooked techniques. Table 1 presents the number of relevant techniques retrieved from each source. It is worth noting that the "Other" category includes many significant techniques that may not have been published in research papers, such as Peach and AFL.

It is still possible for our search to not completely cover all the related papers, but we are confident that the overall trends in this paper are accurate and provide a fair picture of the state-of-the-art techniques.

**Table 1.** Publishers and number of relevant techniques.

| Publisher | Relevant Techniques |
| --- | --- |
| IEEEXplore digital library | 17 |
| ACM digital library | 5 |
| USENIX | 5 |
| Springer online library | 4 |
| Elsevier ScienceDirect | 3 |
| Other | 15 |

## 3. Classification of Network Protocol Fuzzing Techniques

### 3.1. Test-Case-Generation-Methods-Based Classification

In network protocol fuzzing techniques, test cases primarily refer to protocol data packets that are correctly formatted but contain erroneous content. Different network protocol fuzzing techniques employ various methods to generate test cases, and test cases are then sent to the protocol implementation under test using mechanisms such as sockets to identify vulnerabilities. Among the classification criteria for network protocol fuzzing techniques, the generation method of test cases is one of the most important factors. Test case generation methods can be broadly categorized into mutation-based and generation-based approaches, as outlined below:

1. Mutation-Based Approach: In this approach, the fuzzer initially obtains some valid data with proper formatting and content. It then modifies these data using different methods to create corresponding semi-valid data. The mutation process generally involves four methods: bit flipping, arithmetic mutation, block-based mutation, and dictionary-based mutation [14]:

   - Bit flipping involves flipping specific bits within the data packet, changing 0 s to 1 s and 1 s to 0 s.
   - Arithmetic mutation selects a byte sequence, treats it as an integer, performs arithmetic operations, and generates a new integer value, which is then inserted back into the original byte sequence.
   - Block-based mutation treats a given length of byte sequence as a block, which is considered the fundamental unit of the data packet. Operations such as adding, deleting, replacing, and adjusting the priority of blocks are performed.
   - Dictionary-based mutation focuses on specific semantic statements that contain weighted fields. It replaces the weights or other relevant fields with predefined numbers or strings.

2. Generation-Based Approach: In this approach, the fuzzer generates semi-valid data based on known specifications or templates. These templates can be defined by testing personnel themselves or are predefined within the fuzzer.

A comparison of the advantages and disadvantages of these two methods in the context of network protocol fuzzing is given as follows:

1. The mutation-based approach faces challenges because network protocol data packets often contain multiple data types, and different protocols or types of packets have varying specifications. It becomes difficult to find an appropriate mutation strategy to generate test cases that can be correctly received during the test case generation phase. If a random mutation strategy is adopted, significant efforts are required to verify that these test cases can be correctly received.

2. On the other hand, the generation-based approach has its challenges. One of the issues is the cost involved in acquiring the network protocol specifications during the seed acquisition phase. It may require significant resources and efforts to obtain the detailed specifications of the network protocol. Furthermore, the quality of the acquired seed can be compromised if there are deviations in the testing personnel's understanding of the protocol. An inaccurate or incomplete understanding of the protocol can lead to the generation of flawed or ineffective test cases, thereby hindering the effectiveness of the fuzzing process.

### 3.2. Testing-Condition-Based Classification

Based on the level of understanding of the protocol implementation (which refers to the application/software or hardware processes that implement network protocols and handle the sending or receiving of protocol messages), network protocol fuzzing can be classified into three categories: black-box, white-box, and gray-box fuzzing.

1.  Black-box: Black-box fuzzing, also known as random testing, involves the fuzzing tool having no knowledge of the internal workings of the SUT. It can only observe the inputs and outputs of the system to infer its behavior. As a result, black-box fuzzing tends to have lower code coverage compared to other approaches.
2.  White-box: White-box fuzzing requires understanding the internal logic of the target program [17]. The fuzzing tool collects and analyzes information about the internal workings of the system to generate test cases. In the context of network protocol fuzzing, this entails understanding the specific code and runtime information of the protocol implementation. This approach was initially proposed by Godefroid et al. in 2008 to address the limitations of black-box fuzzing in terms of blind and random testing [18,19]. In theory, white-box fuzzing can cover all the code paths in the SUT. However, achieving 100% code coverage is still challenging, especially for large-scale protocol implementations.
3.  Gray-box: Gray-box fuzzing falls between black-box and white-box fuzzing. It adjusts the test case generation method based on dynamic information obtained from the SUT, such as code coverage, branch conditions, and memory states. It aims to generate test cases that cover more execution paths or discover errors more efficiently, without requiring specific knowledge of the code implementation.

In the context of network protocol fuzzing, it is often challenging for testing personnel to access the source code of the protocol implementation. Therefore, black-box and gray-box fuzzing techniques are more commonly used, while white-box fuzzing methods are relatively limited. Compared to black-box fuzzing, gray-box fuzzing has the advantage of adjusting the testing direction based on the obtained protocol implementation information, thereby addressing the issues of blind and random testing encountered in black-box fuzzing.

## 4. Development Timeline of Network Protocol Fuzzing Techniques

Since the application of fuzzing techniques in the field of network protocol security testing in 2001, the development of network protocol fuzzing techniques has spanned approximately 20 years. In this section, based on a timeline and testing conditions, we review and analyze the development of network protocol fuzzing techniques from the aspects of test case generation methods and testing conditions. The goal is to outline the development trajectory and future trends of this technique. The specific analysis of relevant work is depicted in Figure 2.

It can be observed from Figure 2 that, prior to 2017, the majority of network protocol fuzzing techniques employed generation-based black-box fuzzing methods. However, in the past five years, grey-box fuzzing techniques have experienced rapid development. Additionally, considering factors such as the automation level of test case generation steps and the target of the fuzzer, the development of network protocol fuzzing techniques can be categorized into three stages: the initial stage (2001–2009), refinement stage (2009–2017), and development stage (2017–present).

### 4.1. Initial Stage

The initial stage of network protocol fuzzing techniques began with the emergence of these techniques in 2001 and lasted until around 2009. During this stage, the testing tools were mainly general protocol fuzzing frameworks, and employed black-box fuzzing techniques. These fuzzing frameworks either relied on manually constructing test cases by testing personnel or utilized generation-based approaches guided by protocol specifications for test case generation.

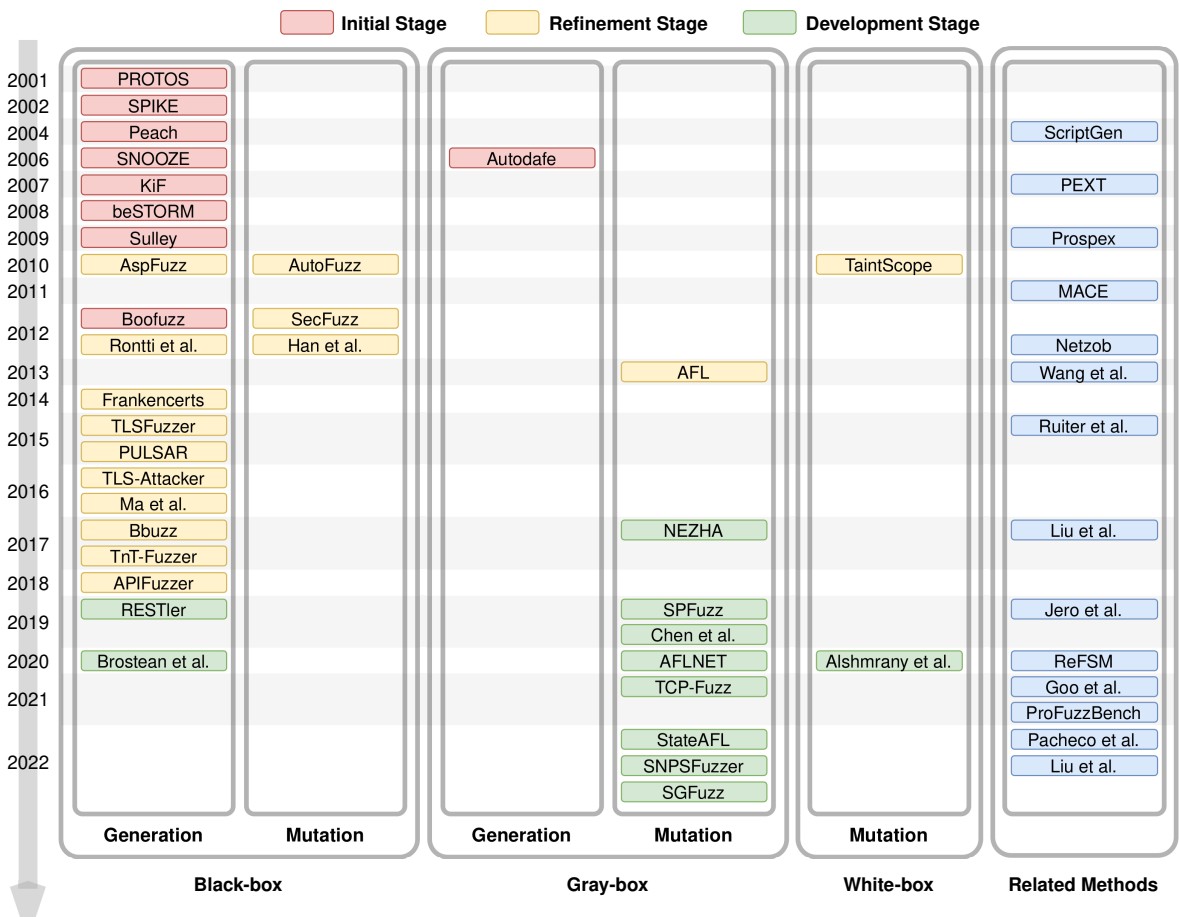

**Figure 2.** Related works of network protocol fuzzing techniques.

### 4.1.1. Work Introduction

1.  Black-box

In 2001, a protocol test suite research project named PROTOS [20] by the University of Oulu introduced the application of fuzzing techniques in the security testing of network protocols. Its aim is to uncover vulnerabilities such as buffer overflows and string format errors. This approach introduces the concept of test suites, which involves manually constructing test messages. These messages are based on the analysis of protocol specifications, considering the supported data structures and the range of acceptable values ranges for each field. However, PROTOS has limitations as it does not provide an API for building custom fuzzing and does not allow for variations in test cases without altering the protocol syntax itself.

In 2002, the SPIKE framework [21], a general-purpose protocol fuzzing framework based on the C language, was introduced. It provides a fuzzing database containing various malformed characters such as long strings, strings with format specifiers, large integers, and negative numbers. SPIKE also offers a rich set of APIs and introduces the concept of block-based protocol fuzzing. Testing personnel using SPIKE can perform fuzzing directly using the provided utility scripts or create their own fuzzers by leveraging the lightweight encapsulation of APIs provided by the framework. However, SPIKE still requires prior knowledge of the protocol for constructing test cases and relies on manual tuning. Additionally, the provided block abstraction is relatively low-level. This makes it difficult to model stateful protocols and complex messages easily.

In 2004, Deja vu Security released the cross-platform fuzzing framework called Peach [22]. It consists of components such as Datamodel (including data types and mutator interfaces), Statemodel (including Datamodel interfaces, states, and actions), proxies, a test-

ing engine, etc. Fuzzers are created by manually writing Peach Pit configuration files. The initial version of Peach was developed in Python and the second and third versions were subsequently released in 2007 and 2013, respectively. The third version was redeveloped in C# and supported the fuzzing of file formats, ActiveX, network protocols, APIs, and more.

In 2006, Greg Banks et al. developed a network protocol fuzzing tool named SNOOZE [23]. It introduces a scenario-based fuzzing approach that allows testing personnel to describe stateful operations in the protocol and generate scenarios consisting of messages generated in each state. The tool generates corresponding test cases based on the fuzzing scenarios, thereby enabling stateful protocol testing. SNOOZE also provides fuzzing primitives targeting specific attacks, thereby allowing testing personnel to focus on specific types of vulnerabilities. The tool can discover vulnerabilities such as buffer overflows, integer overflows, and SQL injections. However, one limitation is that the states observed in the fuzzer are not synchronized with the states in the SUT. This requires further analysis by the testing personnel.

In 2007, H. J. Abdelnur et al. developed the tool KiF [24], which was the first SIP fuzzer that did not solely generate random data. KiF generates test cases based on scenarios and protocol specifications for the SIP. It is capable of automated attacks through self-improvement and tracking the state of the target device. The implementation of KiF relies on a learning algorithm that trains an attack automaton using real network traces. This attack automaton can evolve and update during the fuzzing phase.

In 2008, Beyond Security released the commercial fuzzing tool beSTORM [25]. This tool transforms the BNF used in RFC documents into an attack language, converting protocol specifications into an automated test suite. The architecture of beSTORM consists of two parts: the client side and the monitor side. The client side sends semi-valid packets to the SUT, while the monitor side monitors the state of the SUT, records any exceptions, and sends them back to the client side.

In 2009, the fuzzing framework Sulley was released on the GitHub platform [26]. It is composed of a customizable fuzzer and multiple extensible components. Its advantages include simplifying data transmission, representation, and target monitoring processes. The specific features of Sulley include: (1) monitoring network communication and maintaining relevant records, (2) detecting and monitoring the running status of the target, with the ability to restore to normal operation using various methods, (3) detecting, tracking, and categorizing crashes, (4) conducting fuzzing in parallel, and (5) automatically determining the sequence of test cases that trigger errors. Sulley implements fuzzing by breaking down the protocol requests to be fuzzed into blocks, and then linking the decomposed requests into a session and attaching available monitoring proxies before conducting the testing. Sulley is no longer actively maintained, and Boofuzz [27], released in 2012, is a branch and continuation of the Sulley framework. Boofuzz fixed issues found in Sulley and improved its scalability.

2. Gray-box

In 2006, Vuagnoux introduced Autodafe [28], a fuzzing tool designed to discover buffer overflow vulnerabilities. Autodafe analyzes how user-controlled variables are used by the SUT and prioritizes testing using variables that are passed as parameters to security-sensitive functions. The advantage of Autodafe is its ability to automatically generate protocol descriptions and conduct testing using a fuzzing database.

4.1.2. Summary

Table 2 provides an overview of representative techniques. It can be observed that, as the technique progressed, the initial techniques were limited to testing stateless protocols, while the later techniques supported the testing of stateful protocols. In this stage, fuzzing techniques heavily rely on manual intervention in the process of constructing test cases and have certain limitations in testing flexibility. However, fuzzing tools such as Peach, Sulley, and Boofuzz are still widely used today.

**Table 2.** A summary of representative techniques in the initial stage.

| Tool | Test Case Generation Methods | Testing Conditions | State Support | Key Contributions |
|---|---|---|---|---|
| PROTOS | Generation | Black-box | Stateless | First tool for network protocol fuzzing. |
| SPIKE | Generation | Black-box | Stateless | Introduces block-based strategy for protocol testing. |
| Peach | Generation/Mutation | Black-box | Stateful | Wide protocol coverage, supports testing of files. |
| Sulley | Generation | Black-box | Stateful | Supports parallel fuzz testing. |

*4.2. Refinement Stage*

4.2.1. Work Introduction

1.　　Black-box

In 2010, Gorbunov and Rosenbloom proposed an extensible open-source framework called AutoFuzz [29] for testing network protocol implementations. The framework captures the communication between the client and server to construct a finite-state automaton that learns the protocol implementation. By applying knowledge from bioinformatics, AutoFuzz learns the syntax of individual protocol messages, including message fields and possible types. Using the finite-state automaton as a guide, the framework intelligently mutates the communication session between the client and server to perform testing. During this process, AutoFuzz logs all actions for review and verification by testing personnel. However, AutoFuzz lacks an analysis and comparison between the real feedback from protocol implementations and the ideal feedback from the protocol state machine to determine if specific types of unexpected behavior occur. In the same year, Kitagawa et al. proposed AspFuzz [30], a state-aware fuzzer based on the application-layer protocol specification. Previous message-level fuzzers considered only variations within individual messages without changing the order of message transmission. Scenario-based fuzzers such as SNOOZE and KiF can determine the message transmission order based on scenarios to avoid problems encountered by message-level fuzzers, but the scenario creation process is complex. AspFuzz employs a state-aware approach that allows for selecting the order of test case transmission after generating the test cases. Test cases can be transmitted in the correct order or in an incorrect order. However, AspFuzz relies on a manual definition of the tested protocol and manual determination of successful attacks during the testing process.

In 2012, Tsankov et al. introduced the tool SecFuzz [31] to address the issue of encrypted content in protocol messages of certain protocol implementations. SecFuzz provides the fuzzer with the necessary keys and encryption algorithms to correctly mutate encrypted messages based on network protocol fuzzing. By acting as a middleman between the client and server to intercept messages, SecFuzz obtains valid inputs and classifies them based on three custom fuzz operators before mutating them and forwarding them to the SUT. In the same year, Rontti et al. studied the next-generation network (NGN) fuzzer [32], which creates test cases using protocol specifications. The NGN network refers to a network that integrates all types of services and media. The tool enhances the protocol description using grammar rules derived from the protocol specification and selects the appropriate level of anomalies from an existing anomaly library to generate corresponding test cases. The experimental results demonstrate that the fuzzer is effective in discovering vulnerabilities that can be exploited in DoS or DDoS attacks. Han et al. proposed a relationships analysis and testing case marking (RATM) model-based fuzz testing method for multi-domain fuzzing datasets [33]. By analyzing the relationships between domains in the protocol, the method can directly mutate the corresponding data packets that may trigger vulnerabilities. It can also analyze test results and modify RATM parameters to improve the quality of test cases. This method is highly effective for fuzzing protocols based on the MAC layer.

In 2014, Brubaker et al. designed, implemented, and applied the first large-scale testing tool, Frankencerts [34], specifically targeting certificate verification logic in SSL/TLS

protocol implementations. This tool addresses two main issues: generating high-quality test cases and verifying the reasonableness of accepting/rejecting certificates. Regarding test case generation, Frankencerts creates a corpus containing a massive number of certificates. During test case generation, it combined manually constructed parts of certificates with randomly combined parts of real certificates, ensuring that the generated test cases have a well-formed syntax that can be processed by the protocol. These test cases also violate constraints and dependencies that valid certificates must satisfy, increasing the likelihood of triggering protocol vulnerabilities. Concerning the reasonableness of certificate results, the tool employs differential testing, using multiple independent implementations for joint verification to ensure that the reasons for accepting or rejecting certificates were correct. If the results from most implementations are inconsistent, it indicates that the result is incorrect.

Similarly focusing on TLS protocol implementation validation, in 2015, TLSFuzzer [35], a fuzzing tool for the TLS protocol, was released on GitHub. Unlike typical fuzzers that only check if the SUT crashes, TLSFuzzer also verifies if the system returns the correct error messages. This tool validates the behavior of servers in the TLS protocol, checking if the signature on TLS messages matches the certificate information sent by the server without performing any checks on the protocol certificates. In the same year, Gascon et al. introduced PULSAR [36], a stateful black-box fuzzing tool for proprietary network protocols. This tool is applicable in scenarios where there are no protocol implementation code and protocol specification available. PULSAR integrates fuzzing techniques and automatic protocol reverse engineering to automatically infer the network protocol model based on a set of network data packets generated by the program. The learned network protocol model guides the fuzzing process. The drawback of PULSAR is that the learned model from network messages may naturally lack some functionality, which may affect the testing process of the protocol.

In 2016, Somorovsky et al. developed an open-source framework called TLS-Attacker [37] for evaluating the security of TLS libraries. The framework was implemented using the Maven project management tool. Building upon TLS-Attacker, the authors further proposed a two-stage fuzzing method to evaluate TLS server behavior, automatically detecting unusual padding oracle vulnerabilities and overflows/over-reads. They also established a test suite related to the TLS protocol. In the same year, Ma et al. presented a method for generating fuzzing data using rule-based state machines and stateful rule trees [38]. This method utilizes state machines as formal descriptions of network protocol states, simplifies the state machine by removing known secure paths using protocol rules, and describes the relationships between states and messages using stateful rule trees. Various generation algorithms are employed to regularly mutate initial seeds using data generation algorithms, enabling the generation of fuzzing data.

In 2017, Blumbergs et al. introduced the tool Bbuzz for analyzing network protocols [39], applied specifically in a military context. This tool operates at the bit level, filtering and storing required packet data in files. It assists in identifying field properties by calculating Shannon entropy, enabling an analysis of which parts can be mutated. The reverse engineering results of binary protocols became more accurate as a result.

For representational state transfer (REST) APIs, two fuzzing tools were released on GitHub in 2017 and 2018: TnT-Fuzzer [40] and APIFuzzer [41]. Both tools were written in Python and utilized Swagger specifications to parse HTTP requests, guiding the fuzzing process.

2.   Gray-box

In 2013, Zalewski et al. proposed a mutation-based general fuzzing tool called American Fuzzy Lop (AFL) [42]. This tool introduces code coverage-guided fuzzing by utilizing source code compilation instrumentation and QEMU mode. However, AFL is more suitable for testing stateless projects, such as testing files, and lacks knowledge of the state information of protocol implementations and the structure or order of messages to be sent when

testing network protocols. The process of message mutation is random, thereby resulting in a lower testing efficiency.

3.  White-box

In 2010, Wang et al. presented TaintScope [43], a fuzzing tool that bypasses checksum and validation. This tool employs fine-grained taint analysis to identify inputs that flow into critical system calls or API calls. It also introduces a checksum-aware fuzzing technique that identifies checksum-testing instructions through taint analysis and modifies the SUT to bypass checksum validation. If the modified SUT crashes, TaintScope further repairs the checksum field using hybrid symbolic execution and conducts testing on the original SUT. Additionally, TaintScope monitors how the SUT accesses and uses input data and directionally mutates sensitive information. However, TaintScope has limitations in handling security-related integrity checking schemes, such as digital signatures, and it exhibits lower efficiency when dealing with encrypted data. Moreover, it requires high-quality inputs, depending on both well-formed and malformed input formats.

### 4.2.2. Summary

Table 3 provides a summary of representative techniques in the refinement stage. From the table, it can be observed that, in the refinement stage, fuzzing techniques address the most significant issue present in the initial stage, which is an excessive reliance on manual intervention. For example, Autofuzz and PULSAR could automatically generate protocol state machines by analyzing protocol communication data packets. Additionally, in the refinement stage, fuzzing tools gradually shift their focus from general protocols to specific protocol families. These tools are also capable of addressing specific scenarios during testing. For instance, tools such as Frankencerts and TLSFuzzer target SSL/TLS protocol implementations, SecFuzz focuses on testing encrypted content, and TaintScope bypasses the integrity testing process of the SUT.

On the other hand, during this period, AFL was released as a general fuzzing tool. Although it does not perform well in protocol fuzzing, AFL provides a new approach for protocol fuzzing: using gray-box fuzzing to gather coverage-guided information and improve the efficiency of network protocol fuzzing.

**Table 3.** A summary of representative techniques in the refinement stage.

| Tool | Test Case Generation Methods | Testing Conditions | State Support | Key Contributions |
|---|---|---|---|---|
| AutoFuzz | Mutation | Black-box | Stateful | Automatic generation of state machines. |
| SecFuzz | Mutation | Black-box | Stateful | Enables fuzzing of encrypted data by providing encryption algorithms and keys to the fuzzer. |
| Frankencerts | Generation | Black-box | Stateless | First fuzzer for TLS certificates. |
| PULSAR | Generation | Black-box | Stateful | Automatic generation of state machines. |
| TaintScope | Mutation | White-box | Stateful | First protocol fuzzer to adopt white-box fuzzing approach. |
| AFL | Mutation | Gray-box | Stateful | Adoption of coverage feedback to guide the testing process, serving as a basis for the development of subsequent tools. |

### 4.3. Development Stage

The development stage, spanning from 2017 to the present, is characterized by the dominance of gray-box fuzzing techniques that leverage code coverage to improve the efficiency of fuzzing.

### 4.3.1. Work Introduction

1.  Black-box

In 2019, Atlidakis et al. developed the first stateful fuzzer for REST APIs, called RESTler [44]. This tool analyzes the OpenAPI specification of cloud services to extract REST

syntax and infer dependencies between different types of requests. RESTler employs three different search strategies based on dynamic feedback from service responses to assist in the generation of test cases. Additionally, RESTler introduces bucketing schemes to cluster similar vulnerabilities and aid users in vulnerability analysis.

In 2020, for the Datagram Transport Layer Security (DTLS) protocol, Brostean et al. extended the TLS-Attacker framework and built a stateful fuzzing framework for DTLS servers [45]. The tool used model learning with the TTT algorithm to infer Mealy machines and performed comprehensive protocol state fuzzing on DTLS. The experiments analyzed Mealy state machine models of 13 DTLS implementations, revealing 4 severe security vulnerabilities.

2. Gray-box

In 2017, Petsios et al. modified the LibFuzzer framework [46] and proposed an efficient differential testing tool called NEZHA [47]. NEZHA introduces the concept of $\delta$-diversity to capture behavioral inconsistencies among multiple SUTs. NEZHA consists of runtime components and a core engine: the runtime components collect and transmit all necessary information for $\delta$-diversity guidance to the core engine, which generates new inputs through mutation to uncover differences between SUTs and update the seed corpus guided by $\delta$-diversity. During testing, NEZHA determines whether to employ black-box or gray-box methods based on whether the SUT supports instrumentation or binary rewriting.

In 2019, Song et al. proposed SPFuzz [48], a stateful protocol fuzzing framework that aims to build a flexible and coverage-guided approach. SPFuzz combines the language specification from Boofuzz to describe protocol specifications, state transitions, and dependencies for generating valuable test cases. It maintains correct messages in the session state and handles protocol dependencies by updating message data in a timely manner. SPFuzz employs a three-level mutation strategy (headers, content, and sequences) and incorporates a random allocation of messages and weights for mutation strategies to cover more paths during the fuzzing process. One limitation of SPFuzz is that it requires the source code and specification of the protocol, with the protocol specification relying on manual construction. In the same year, in order to achieve higher code coverage in testing network communication protocols, Chen et al. designed a stateful protocol fuzzing strategy and demonstrated the limitations of stateless gray-box fuzzers in protocol testing [49]. The stateful fuzzing strategy consists of a state transition engine and a multi-state fork server. It performs a search on different fuzzing states using a depth-first search algorithm and determines the progression and regression of states flexibly through energy scheduling. This strategy allows for the flexible fuzzing of different states in protocol programs.

In 2020, Pham et al. developed AFLNET [50], which uses response codes as states for network protocol programs. AFLNET accurately fuzzes the actual states of network protocol programs and utilizes state feedback to guide the fuzzing process. AFLNET uses a mutation-based approach and uses the message exchange sequence between clients and servers as initial seeds. During testing, AFLNET acts as a client and replays variants of request message sequences. It also applies coverage-guided techniques to retain variants that effectively improve code or state coverage. AFLNET shows significant improvements compared to coverage-guided stateless testing tools such as AFLnwe (a network-enabled version of AFL) and the generation-based testing tool Boofuzz. However, AFLNET is not suitable for protocols without state codes, and the extraction of state information relies on manually written protocol specifications.

In 2021, Zou et al. designed TCP-Fuzz [51], a novel fuzzing framework for effectively testing TCP stacks and detecting errors within them. TCP-Fuzz adopts a dependency-based strategy to generate effective test cases by considering dependencies between system calls and packets to generate sequences of system calls. To achieve an efficient coverage of state transitions, TCP-Fuzz employs a transition-guided fuzzing method that utilizes a new code metric called branch transition as program feedback instead of code coverage. The branch transition is represented as a vector that stores the branch coverage of the current input (packet or system call) and the changes in branch coverage between the current

input and previous inputs. This approach not only describes states but also captures state transitions between adjacent inputs. Lastly, in order to detect semantic errors, TCP-Fuzz uses a differential checker that compares the outputs of multiple TCP stacks with the same input. Since different TCP stacks should adhere to many of the same semantic rules (most of which are defined in RFC documents), inconsistencies in output indicate possible semantic errors in some TCP stacks.

In 2022, Natella developed StateAFL [52], a gray-box fuzzer for network servers. StateAFL performs compile-time instrumentation to detect the target server and insert probes for memory allocation and network I/O operations. At runtime, the fuzzer captures snapshots of long-lived memory regions and applies a locality-sensitive hashing algorithm to map the memory content to unique state identifiers. This allows StateAFL to infer the current protocol state of the SUT and gradually build a protocol state machine to guide the fuzzing process. Qualitative analysis shows that inferring states from memory provides a better reflection of server behavior than relying solely on response codes. In the same year, Li et al. proposed another fast gray-box fuzzer called SNPSFuzzer [53], which also uses snapshots. SNPSFuzzer builds upon AFLNET and introduces three main components: a snapshot-based instance generator, a snapshotter, and a message chain analyzer. When the network protocol program reaches a specific state, contextual information is stored, and it can be restored when needed. Additionally, Li et al. designed a message chain analysis algorithm that splits the message chain into prefix, infix, and suffix information using two variables for analysis. This approach explores deeper network protocol states. Compared to AFLNET, SNPSFuzzer achieved a 112.0% to 168.9% speed improvement in network protocol fuzzing within 24 hours and increased path coverage by 21.4% to 27.5%. In addition, in 2022, na et al. developed SGFuzz [54], a tool that automatically analyzes protocol states and performs stateful testing by leveraging the regularity between variable names in protocol implementations. SGFuzz uses pattern matching to identify state variables using enums. When a state variable is assigned a new value, the tool sends a corresponding notification and adds the new state to the constructed state transition tree (STT). SGFuzz also adds generated inputs to the seed library for training the STT and focuses on nodes that are rarely visited and have descendants that are more likely to traverse different paths, thereby improving the coverage of the state space. SGFuzz achieves significant improvements in generating state sequences, achieving the same branch coverage and discovering stateful errors in the absence of explicit protocol specifications or manual annotations. However, SGFuzz requires the protocol implementation's source code for analyzing protocol states during the testing process, and the analysis of vulnerabilities triggered by hidden states still requires manual analysis.

3.    White-box

In 2020, Alshmrany et al. proposed a verification method that combines fuzzing with symbolic execution techniques [55]. The approach is based on AFL and utilizes fuzzing for the initial exploration of network protocols. Simultaneously, symbolic execution is employed to explore program paths and protocol states. By combining these techniques, high-coverage test case data packets can be automatically generated for network protocol implementations. The symbolic execution is implemented using both path exploration and bounded model checking (BMC) methods.

4.3.2. Summary

Table 4 summarizes the representative techniques in the development stage. In this stage, several gray-box fuzzing techniques were developed based on AFL, adapting AFL's stateless fuzzing to stateful fuzzing suitable for protocols. AFLNET, StateAFL, SNPSFuzzer, and SGFuzz all utilize state feedback to guide the fuzzing process. They achieve this by recording message response codes, capturing memory snapshots, storing contextual information, and analyzing protocol source code to identify specific data types and determine protocol states. Alshmrany et al. [55]employed symbolic execution techniques to analyze program paths and explore protocol states.

In general, during this stage, the focus is on fuzzing stateful protocols, and the main goal of these fuzzing tools is to uncover vulnerabilities in deeper protocol states.

**Table 4.** A summary of representative techniques in the development stage.

| Tool | Test Case Generation Methods | Testing Conditions | State Support | Key Contributions |
|---|---|---|---|---|
| NEZHA | Mutation | Gray-box | Stateless | Guides seed library updates using $\delta$-diversity. |
| SPFuzz | Mutation | Gray-box | Stateful | Uses code coverage as feedback to improve fuzzing efficiency. |
| AFLNET | Mutation | Gray-box | Stateful | First fuzzer to use state-aware feedback. |
| StateAFL | Mutation | Gray-box | Stateful | Represents protocol state through memory snapshots. |
| SGFuzz | Mutation | Gray-box | Stateful | Exploits the tendency of existing protocol implementations to name states with specific names. |

## 5. Related Methods of Network Protocol Fuzzing Techniques

In addition to the network protocol fuzzing tools/frameworks mentioned above, there are other related tools and methods that contribute to the automation and evaluation of network protocol fuzzing.

### 5.1. Automatic Generation Techniques of Protocol State Machines

Finite state machines (FSMs) are often used in the field of network protocol research to model the message exchange process of network protocols in order to describe the state transitions of protocol entities. In order to construct fuzzy test cases for unknown protocols, it is necessary to utilize the protocol specifications extracted through protocol reverse engineering. For stateless protocols, test cases can be generated based on the protocol format to ensure that each field breaks through the verification of the target program. However, for stateful protocols, it is necessary to send test message sequences that can be accepted by the protocol state machine in a targeted manner in order to avoid a large number of invalid test cases due to state mismatches and to ensure the depth and efficiency of testing. Traditional fuzz testing does not include context information and all states in the message sequence, so the test data generated for each state are discrete and may not cover the entire state trajectory. Therefore, vulnerabilities in state transitions may go undetected, and there may be a large amount of redundant test data. Test data are randomly generated and lack rules. Furthermore, protocol fuzzing techniques based on generation can utilize protocol state machines to construct protocol templates. Therefore, combining finite state machine generation technology can make protocol fuzz testing more targeted and efficient, providing better test coverage.

Next, we will introduce the development history of automatic protocol state machine generation technology.

#### 5.1.1. Passive Inference-Based Approaches

Passive inference refers to the process of inferring the state machine from a given set of finite samples without relying on guidance from the protocol entity. It primarily involves two phases: state labeling and state machine simplification. In the state labeling phase, classification labels can be assigned based on features such as message types, lengths, and positions in the sample data, resulting in different states. In the state machine simplification phase, the labeled states can be further simplified to generate a concise state machine for subsequent analysis and applications.

In 2004, the first analysis tool designed to infer protocol state machines from network data streams, called ScriptGen [56], was proposed. This tool uses the Needleman–Wunsch sequence alignment algorithm similar to the PI project, as well as the micro-clustering and macro-clustering of captured network traffic to construct protocol state machines. However, this tool has certain limitations, such as its inability to handle different message sessions with causal correlations and its reliance on TCP packet identifiers (e.g., ACK, SYN, FIN) as

priors to address issues related to TCP packet reassembly, retransmission, and out-of-order packets. As a result, it can only analyze a few specific protocols and is not a robust and widely applicable solution for inferring protocol states.

In 2007, Shevertalov et al. initially proposed the solution named PEXT [57], which is based on packet clustering for automatic protocol state machine inference. This solution calculates the distance measure D(a,b) between packets a and b based on the length of their longest common subsequence. It then clusters packets based on D(a,b) and annotates packets with the same source and destination addresses as initial state transition sequences. It subsequently merges state pairs with common prefixes and no sibling nodes to obtain the state machines for the entire sample set. However, PEXT has two limitations: (1) it does not fully consider the role of keyword fields in state machine transitions, leading to less accurate and precise state labeling, and (2) the merging process is too simple, resulting in less concise inferred state machines.

In 2009, Comparetti et al. proposed Prospex [58], a solution that infers protocol states based on binary executable code. This tool improves upon their previous work [59] on behavior-based message format extraction. It introduces message structures and the impact of messages on server behavior. During the session analysis phase, Prospex uses dynamic taint analysis to track all operations involving data reads from protocol messages. It splits the session into messages, considering the first input byte received by the server as the start of a message and treating all subsequent inputs as part of that message until the server sends a reply. This process is repeated until all tracked data packets are segmented into messages. This approach is more accurate than treating each packet as a message, particularly for interaction-based protocols where a message may span multiple data packets. It then extracts features from sample packets, clusters them to obtain a set of message types (M), represents session samples (S) as message-type sequences (MTSs), and constructs the augmented prefix tree acceptor (APTA) using S. In the APTA, nodes represent states, and edges represent inputs $a_i \in M$ causing state transitions. The predecessor types $P_i$ for each type $m_i \in M$ are also deduced. Finally, the Ex-bar algorithm is used to merge identical states and extract a minimal deterministic finite automaton (DFA). However, according to research [60], inferring an accurate minimal state machine solely based on positive samples captured from the network is difficult.

### 5.1.2. Active-Inference-Based Approaches

Passive inference algorithms rely on the completeness of the sample set. To address this limitation, research on active inference began with the introduction of the L* algorithm by Angluin et al. [61]. In active inference, the goal is to expand the original sample set using a manual learning system to iteratively infer the state machine.

The L* algorithm divides input–output examples into two sets: positive samples and negative samples. Positive samples are input sequences that the automaton can handle correctly, while negative samples are input sequences that the automaton cannot handle correctly. The L* algorithm assumes the existence of an Oracle that can provide accurate answers. There are two types of queries: member query and equivalence query. This algorithm uses these examples to construct a hypothesis set OT and generates all candidate state machines M. As the algorithm progresses, M is evaluated against the true state machine using equivalence queries to gradually narrow down the hypothesis set and find the minimized state automaton. The L* algorithm ensures that a complete minimal state machine can be inferred in polynomial time, but the key challenge is how to accurately answer the queries.

In 2011, Cho et al. proposed the tool MACE based on the L* algorithm [62]. MACE relies on dynamic symbolic execution to discover protocol messages and uses a special filtering component to select messages for learning models. It guides the further search using the learning model and refines it when new messages are discovered. These three components alternate until the process converges, automatically inferring the protocol state machine and exploring the program's state space. MACE can infer protocol models

and explore the search space of programs, generating tests automatically. In an experimental analysis on four programs, MACE discovered seven vulnerabilities and achieved good results.

In 2012, Bossert et al. introduced the open-source project Netzob [63], which consists of three modules: lexical inference module, syntactic inference module, and simulation module. The lexical inference module adopts the multiple sequence alignment algorithm from PI [64] and improves upon the L* algorithm. It uses the feature information in message formats to infer Mealy machines. The inferred lexical and syntactic specifications are then used in the simulation module to simulate communication between protocol entities, enabling intelligent fuzzing for unknown protocols.

In 2013, Wang et al. proposed and designed the Veritas system [65]. Veritas builds a probabilistic protocol state machine by using a set of protocol state information obtained through cluster analysis. For each protocol state information, Veritas measures its frequency of occurrence and the transition probability between it and other protocol state information. Veritas then constructs a labeled directed graph to represent the protocol state machine using these statistical results, with state transitions and transition probability values as labels on the directed edges. Finally, Veritas transforms the directed graph into a probabilistic protocol state machine. To reduce the complexity of the protocol state machine, Veritas uses the Hopcroft–Karp algorithm to perform minimization operations on the protocol state machine. The Hopcroft–Karp algorithm merges equivalent states in the protocol state machine into a single state, reducing the number of states and improving the efficiency and readability of the protocol state machine.

In 2015, De Ruiter et al. proposed a method to describe protocols using state machines [66]. They employed an improved version of the L* model learning algorithm to infer the state machine through LearnLib, which provides an abstract input message list (also known as an input alphabet). De Ruiter et al. used a testing tool to transform the input message list into concrete messages sent to the SUT and received responses that were then transformed into abstract message types. LearnLib analyzed the returned message types and made hypotheses about the protocol state machine. Analyzing different TLS implementations produces unique and distinct state machines, indicating that this technique can also be used for TLS fingerprinting. The problem with this approach is that, after obtaining the protocol state machine, De Ruiter et al. manually analyzed the state machine to find logic vulnerabilities in specific implementations and then analyzed the implementation source code to identify corresponding issues, rather than using an automated method to test protocol entities.

In 2017, Liu et al. proposed a technique to address the problem of the excessive generalization of state machines caused by errors in merging labeled states when constructing the APTA tree [67]. They utilized a dynamic taint analysis technique, relying on DECAF to analyze network applications and construct an APTA tree. By using semantic information to differentiate states and merging similar states, they improved the accuracy of labeling states in the APTA tree. The method was tested with TCP and the Agobot control protocol, and it achieved good results.

In 2019, Pacheco et al. studied the automatic learning of protocol rules from textual specifications (i.e., RFC) and applied the automatically extracted protocol rules to a fuzzer for evaluating the learned rules [68]. The evaluation showed that this approach could discover the same attacks as manually specified systems with fewer test cases. In 2022, Pacheco et al. further proposed a more comprehensive method [69]. The method included three key steps: (1) a large-scale word representation learning of technical languages, (2) a zero-shot learning mapping of protocol text to an intermediate language, and (3) rule-based mapping from the intermediate language to specific protocol finite-state machines. They extracted protocol state machines from protocol specifications of TCP, DCCP, BGPv4, and other protocols, achieving good results.

In 2020, LI et al. proposed a protocol state inference method called ReFSM [70], which improved upon existing protocol state inference methods by taking into account

the real-time packet capture feature. This method is based on the extended finite-state machine (EFSM) and consists of three steps: (1) message type identification, which uses the a priori keyword analysis to extract protocol keywords and the K-means algorithm to group messages to determine the number of clusters, with each group being treated as a different message type. (2) EFSM construction and semantic inference, which constructs a tree of protocol transition automata (PTAs) that accepts all protocol sessions and uses the K-tail merge algorithm to simplify the protocol state machine. (3) Sub-data set extraction, which extracts sub-data sets containing message field values observed in the messages, for further analysis to search for correlations between fields in the messages. The accuracy of the extracted state-related fields has been greatly improved by combining the state-related field method with the clustering method. Then, the information of these state-related fields is used to compare each transition in the EFSM, and the merge operation of the state machine is completed by determining whether the generated A and B trees have the same structure. Although the ReFSM method has achieved some improvement in the extraction of state-related fields, there is still a problem of state explosion, which affects the efficiency and accuracy of state machine inference due to a large number of states in the protocol state machine.

In 2021, building on previous research on the $L_M^+$ algorithm for inferring Mealy machines, Goo et al. proposed a new algorithm and a novel solution for inferring protocol states [71]. They modeled the Mealy machine of client–server-type protocols through input and equivalence queries on the input character sequence. In feasibility tests, the tool was tested on the Modbus and MQTT protocols, yielding good results.

### 5.1.3. Summary

In summary, protocol state machines can systematically describe the behavior and state transitions of a system or network protocol, providing clear testing directions for fuzz testing and making fuzz testing more targeted and efficient. By utilizing protocol state machines, designed fuzz testing cases can cover all states and transitions of the tested system or network protocol in terms of structure, thus improving testing coverage.

Currently, it is the most fundamental and important research direction in this field to automatically extract protocol state machines from protocol implementations or network traffic, or directly map protocol behavior to state machines with well-defined states and state transitions. One important development direction is based on active inference, which uses artificial learning systems to continuously expand the original sample set and repeatedly infer state machines, thereby reducing the dependence on sample set completeness and improving the accuracy and efficiency of state machine inference.

### 5.2. Evaluation of Network Protocol Fuzzing Techniques

The number of network protocol fuzzing tools is increasing, and each tool focuses on different testing targets and problem domains. Therefore, to determine which testing tool provides the best results, it is necessary to evaluate network protocol fuzzing techniques.

In 2021, Natella et al. introduced a benchmark test suite for stateful network protocol fuzzing called ProFuzzBench [72]. The benchmark test suite includes a set of representative open-source network servers for popular protocols and automated experimental tools such as AFLNET and StateAFL. The test suite is implemented using Docker to achieve reproducible experiments and supports the comparative analysis of different fuzzing techniques under controlled conditions.

Since not all states in a stateful protocol are equally important, and fuzzing techniques have time limitations, an effective state selection algorithm is needed to filter out good states with higher priority. In 2022, Liu et al. evaluated a set of state selection algorithms using the AFLNET tool on the ProFuzzBench benchmark test suite and proposed an improved state selection algorithm called AFLNETLegion [73].

*5.3. Summary*

In conclusion, when choosing fuzzing tools that require the manual construction of protocol specifications, such as Peach and SNOOZE, automated tools can be selected based on existing protocol-related information to generate protocol state machines and reduce manual effort. On the other hand, when testing personnel need to figure out which methods and fuzzing tools are more effective in specific scenarios, ProFuzzBench supports the testing of representative tools and enables the comparative analysis of different fuzzing techniques under controlled conditions.

## 6. State-of-the-Art Techniques

According to the general process of protocol fuzzing and the difference between traditional fuzzing and protocol fuzzing described in Section 1 and the related methods of protocol fuzzing techniques introduced in Section 5, the following questions should be considered:

1. How to generate or select test cases;
2. How to validate those inputs against the specification of the SUT;
3. How to direct SUT to conduct tests for deeper protocol states;
4. How to improve the automation level of protocol fuzzing.

In this section, we address RQ1 and RQ2 by summarizing and comparing the main contributions to the above issues of protocol fuzzing in today's baseline and state-of-the-art techniques.

*6.1. Test Cases Generation and Selection*

In the test case generation phase, two main approaches can be distinguished: generation-based methods represented by Peach and Boofuzz, and mutation-based methods represented by AFLNet.

In the generation-based approach, such as Peach and Boofuzz, a protocol template is required, which defines the desired packet format for the protocol. Manual effort is involved in obtaining the protocol template, which specifies the structure and expected behavior of the protocol's data packets. Test cases are then generated based on this template, incorporating variations and mutations to explore different input scenarios.

On the other hand, mutation-based methods, such as AFLNet, StateAFL, SNPSFuzzer, and SGFuzz, rely on real network traffic to generate test cases. By capturing and analyzing actual network packets, predefined techniques are applied to mutate the captured packets and generate desired test cases. This approach leverages the existing network traffic to explore potential vulnerabilities and test the robustness of the protocol implementation.

As discussed in Section 3, one of the major challenges with generation-based methods is the reliance on testing personnel possessing prior knowledge of the protocol. It requires manual effort to gather information about the protocol specification, including the states and transition conditions. This approach can result in significant research costs and expertise requirements. In the case of mutation-based methods, a notable challenge is that the current test case generation strategies can be quite random, where many test cases generated through mutation may fail to pass the protocol's validation, leading to a significant impact on the efficiency of mutation-based fuzzing techniques.

*6.2. Input Validation*

The ability of automatically generating numerous test cases to trigger unexpected behaviors of the SUT is a significant advantage of fuzzing. However, if the SUT has an input validation mechanism, these test cases are quite likely to be rejected in the early phase of execution. In the field of protocol fuzzing, checksum and encryption techniques are commonly employed to ensure the integrity and confidentiality of input data packets.

TaintScope first uses dynamic taint analysis and predefined rules to detect potential checksum points, and then mutates bytes to create new test cases and changes the checksum

points to let all created test cases pass the integrity validation. When some test cases can make the SUT crash, it uses symbolic execution and constraint solving to fix the checksum value of these test cases. One of the major challenge with TaintScope is that it requires extensive information about the inputs, including both well-formed and malformed inputs. The quality of the inputs significantly impacts the identification of checkpoints. Therefore, it is crucial to provide a diverse and comprehensive range of inputs, including various formats and error conditions, to ensure thorough testing and an accurate identification of checkpoints by TaintScope.

SecFuzz first address the issue of encrypted content in protocol messages. It provides the fuzzer with necessary keys and encryption algorithms, and mutates encrypted messages by acting as a middleman between the client and server. Currently, SecFuzz is capable of addressing only certain symmetric encryption issues and does not support mutation for all fields.

### 6.3. Stateful Protocol Fuzzing

In the context of protocol implementation in a server–client mode, a server is stateful and message-driven. It takes a sequence of messages from client, handles the messages, and sends appropriate responses, and the server's response depends on both the current message and the current internal server state, which is controlled by earlier messages.

In stateful protocol fuzzing, generation-based and mutation-based methods employ different approaches.

In generation-based methods, the protocol template contains relevant information about the protocol's states. The fuzzer not only generates test cases but also constructs corresponding state machines. In Peach, the protocol state machine is represented using the StateModel format, while, in Boofuzz, it is represented using sessions. A noteble challenge is that manual effort is involved in obtaining the protocol template in both Peach and Boofuzz.

In mutation-based methods, AFLNET pioneered stateful coverage-based graybox fuzzing, integrating automated state model inference and coverage-guided fuzzing. Subsequently, StateAFL, SNPSFuzzer, and SGFuzz have further explored state-feedback-based protocol fuzzing. AFLNET constructs the protocol state machine using status code feedback. During seed generation, if a generated test sequence triggers a new state transition, it is added to the seed corpus, and the new state is incorporated into the state machine. AFLNET faces limitations when protocols do not provide status codes, rendering it ineffective. StateAFL and SNPSFuzzer infer the protocol's state by taking snapshots of the target server and gradually build the protocol state machine to guide the fuzzing process. They maintain interesting seeds and states based on feedback metrics such as state coverage. However, their granularity is at the process level, which may result in state machine discrepancies with the actual protocol state. SGFuzz automatically identifies state variables in program code and captures protocol state changes by constructing a state transition tree. This approach relies on the protocol's source code and is not suitable for the security testing of closed-source or proprietary protocols.

### 6.4. Automated Testing

Section 6.1 mentions that generation-based protocol fuzzing techniques, represented by Peach and Boofuzz, require manual assistance in obtaining protocol templates, and their level of automation is relatively low. Therefore, natural language processing (NLP) techniques can be employed to aid in the extraction of protocol state machines during the protocol templates extraction phase.

In the protocol state machine extraction phase, there are two types of approaches: passive inference and active inference. Passive inference, represented by PEXT, is based on message clustering for protocol state machine inference. Active inference research is mainly based on the L* (learning algorithm) algorithm, such as tools such as LearnLib, Netzob, MACE, etc., assuming the existence of an Oracle that can provide accurate answers to

membership queries and equivalence queries. Firstly, a closed and continuous observation table (OT) is constructed based on membership queries, and then the corresponding candidate state machine M is generated. Afterwards, equivalence queries are used to determine whether M is consistent with the actual state machine. If yes, the inference is terminated; otherwise, a counter example is generated to re-infer. One of the main challenges of the generation-based approach is that its efficiency depends on the number of generated queries and the response speed of the Oracle to the queries. Active inference based on the L* algorithm requires the generation of a large number of queries, resulting in low efficiency.

In recent years, research on protocol state inference has also focused on methods based on deep learning and machine learning techniques [70,74]. However, these methods still have limitations and shortcomings. For example, a large amount of training data is required to train the model, and it may be difficult to obtain sufficient training data for some protocols, making them only suitable for simple protocols.

## 7. Technique Development Context and Future Directions

In this section, we answer RQ3 by discussing some of the possible future directions of the protocol fuzzing and related techniques. Although we cannot accurately predict the future directions that the study of protocol fuzzing will follow, it is possible for us to identify and summarize some trends based on the reviewed papers.

### 7.1. Development Context

As mentioned earlier, AFL has discovered numerous zero-day vulnerabilities in mainstream open-source software since its release in 2013, and it has been widely used and extended for testing different targets. Its success has demonstrated the value of code coverage-guided techniques in practical fuzzing and represents an important role in the development of fuzzing techniques. Analyzing the development trends, we can observe that AFL has had a significant impact on the research focus of network protocol fuzzing techniques.

Referring to Figure 3, we can see that, before the release of AFL, the research focus of testing personnel was on black-box fuzzing techniques. However, after AFL was released, gray-box fuzzing techniques gradually gained attention. Based on these findings, we can speculate on the following development trends for network protocol fuzzing techniques:

1.  As time develops, testing personnel would capture relevant information about protocol implementations more easily and improve testing efficiency based on feedback such as state coverage and code coverage. Consequently, the number of newly released black-box fuzzing tools is likely to decrease gradually, while the number of gray-box fuzzing tools will increase and dominate the network protocol fuzzing landscape.
2.  The number of white-box fuzzing tools may increase because, as general network protocol fuzzing techniques become more mature, specific requirements for network protocol fuzzing will also emerge; for example, fuzzing techniques that bypass encryption keys or integrity checks. Accomplishing these specific testing requirements requires a more comprehensive understanding of the relevant information about protocol implementations, hence the need for white-box fuzzing techniques.

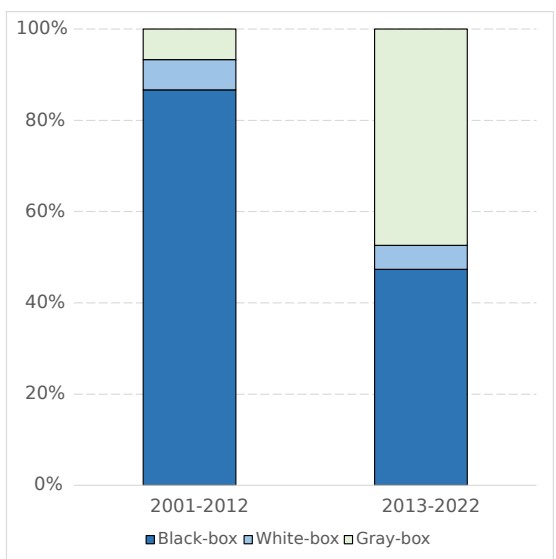

**Figure 3.** The percentage of techniques before and after the release of AFL.

### 7.2. Future Directions

#### 7.2.1. Automation in Generation-Based Fuzzing Techniques

As mentioned before, the analysis of protocol formats in network protocol fuzzing requires significant effort, distinguishing it from other fuzzing approaches. The manual analysis of network protocol formats improves the effectiveness of test cases to some extent, but it incurs substantial research costs. Additionally, analysis needs to be performed again for different protocols, resulting in high manual effort. Consequently, the practical value of manual access to network protocol formats is greatly diminished. Feasible solutions to this problem include:

1. Utilizing NLP and other AI techniques to automatically analyze protocol specification documents and network traces, thereby enhancing the automation level of protocol format analysis and overall fuzzing automation. However, the current implementation of this approach is not yet optimal, and there are still many unresolved issues. For instance, prior to automatic analysis using AI techniques, the manual annotation of protocol documents is required to assist in protocol state machine generation.
2. Large language models (LLMs) are currently a trending area in AI research, as they can better comprehend and generate high-quality text. In the field of network protocol fuzzing, it is worth considering training dedicated LLMs to generate malformed protocol packets. This approach can help to overcome the significant manual effort involved in existing generation-based fuzzing techniques.

#### 7.2.2. Efficiency Improvement in Fuzzing Techniques

In network protocol fuzzing, the effectiveness of test cases is crucial for discovering unknown vulnerabilities. Efficient test cases exhibit high mutation rates, acceptance rates, and code coverage. Optimizing test case selection strategies and ensuring the use of minimal test case sets to uncover as many potential vulnerabilities as possible are hot topics in future research on network protocol fuzzing. Possible solutions to consider include:

1. By altering the mutation strategy, the mutation-based protocol fuzzing technique can generate test cases that better conform to the characteristics of protocol data packets, thereby enhancing the success rate of validation. Mutation strategies encompass various approaches, such as the three-level mutation strategy employed in SPFuzz, context-aware structured mutation strategies, etc.
2. Combining neural network models with network protocol fuzzing techniques to extract rules and knowledge from massive data. The neural network model can automatically filter out invalid test cases, thereby improving the efficiency of fuzzing.

3.  Applying directed fuzzing principles and imposing certain constraints during the test case generation phase to generate test cases that cover target code/state. These target code/state segments are more likely to trigger security events in the protocol. Testing these code/state segments can lead to a higher testing efficiency.

### 7.2.3. Variety of Test Targets in Network Protocol Fuzzing

In current network protocol fuzzing techniques, the test targets mostly adhere to a client/server architecture. The entire testing process involves communication between the fuzzer and the protocol implementation, with the fuzzer sending test cases to the protocol implementation. There is limited support for multi-party protocols or protocols where users have relatively equal positions. Additionally, in terms of network hierarchy, there is a greater focus on testing a simple application-layer protocol such as a file transfer protocol (FTP) and less emphasis on lower-level and more complex protocols.

For lower-level protocols, the corresponding protocol implementations may be non-existent. It is challenging for the fuzzer to observe abnormal behaviors such as crashes in the protocol implementation to determine the discovery of new vulnerabilities. However, for different protocols, specific security events can be defined, and the success of the testing can be assessed indirectly through other network metrics. For example, changes in throughput or modifications in the routing table can be observed to determine whether the fuzzer has discovered vulnerabilities.

### 7.2.4. Related Techniques of Network Protocol Fuzzing

The active learning method based on the L* algorithm is currently widely applied to protocol state machine inference, but it still faces several challenges. One major issue is excessive membership queries, which may lead to incomplete counterexample sample sets and potentially affect the accuracy of the inferred protocol state machine. Additionally, this method may suffer from generalization problems when dealing with complex protocols, meaning it may fail to correctly infer certain behaviors of the protocol state machine. To address these challenges, future research could explore the following areas for expansion and improvement.

First, incorporating the sequential constraint relationships between messages into protocol state machine inference can better reflect the actual behavior of the protocol since messages in a protocol often have a certain degree of temporal order. Second, exploring the structural correlations between counterexamples and positive examples may help to generate counterexamples more effectively and reduce the number of membership queries. Furthermore, optimizing the generation process of queries and counterexamples can improve the performance of the method, such as using more efficient algorithms or strategies to generate queries and counterexamples. Finally, combining the active learning method with other formal methods can improve the accuracy and reliability of protocol state machine inference; for example, using model checking to verify the inferred protocol state machine's correctness or combining the active learning method with other machine learning techniques to improve the efficiency and accuracy of inference.

For the evaluation of protocol fuzzing techniques, Profuzzbench currently only supports code coverage as a metric for comparison, and the included fuzzers do not encompass generation-based fuzzing techniques. In future benchmark test suites, it is recommended to incorporate test suites that involve generation-based fuzzing techniques and evaluate protocol fuzzing techniques based on multiple metrics, including the number of bugs discovered, time required to test the same bug, state coverage, and other relevant indicators.

## 8. Conclusions

Network protocol vulnerability discovery techniques are essential for ensuring secure network communication. Among various vulnerability discovery techniques, fuzzing attracts wide attention thanks to its low deployment complexity and minimal prior knowledge requirements about the SUT.

This paper reviews and summarizes the generation, development, and application of various network protocol fuzzing and related techniques based on a timeline. Starting with the introduction of the background, analysis of the working principles, and classification methods of network protocol fuzzing techniques, we provide an overview of the research progress in this field from the perspectives of white-box, gray-box, and black-box fuzzing techniques. Each of these perspectives includes the introduction of typical tools and methods. Based on the analysis of approximately fifty related papers over the past two decades, the paper summarizes the development patterns and existing issues in network protocol fuzzing techniques, and introduces NLP techniques that can be combined with protocol fuzzing techniques. Furthermore, the paper provides prospects for future research directions in this field, aiming to contribute to the efficient development of research work in this area.

**Author Contributions:** Conceptualization, Z.Z. and J.Z.; methodology, Z.Z. and J.Z.; validation, Z.Z., H.Z., J.Z. and Y.Y.; formal analysis, Z.Z. and H.Z.; investigation, Z.Z. and H.Z.; resources, J.Z.; writing—original draft preparation, Z.Z. and H.Z.; writing—review and editing, J.Z. and Y.Y.; visualization, Z.Z.; supervision, J.Z.; project administration, J.Z. All authors have read and agreed to the published version of the manuscript.

**Funding:** This research received no external funding.

**Data Availability Statement:** The data presented in this study are available on request from the corresponding author.

**Conflicts of Interest:** The authors declare no conflict of interest.

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
