# Peer review of "A Survey on the Development of Network Protocol Fuzzing Techniques"

_electronics, doi:10.3390/electronics12132904_

Round 1
Reviewer 1 Report
The paper is devoted to producing a survey on protocol fuzzing techniques.
The authors selected the criteria based on development maturity (initial stage, refinement stage, and development stage) as well as testing conditions (black-box, grey-box, white-box). Several tools [18-71] have been reviewed and some further predictions for the subject area have been proposed.
In general, for the review type article, the paper is ok. However, the reviewer can state three weak points:
1. Existing surveys are not well checked nor their methods described. Please provide the readers with the methods used in [7,13,14], discuss their conclusions in brief as well as show us the advantages of your approaches that motivated you to write a new survey.
2. The usage of formal methods for protocols (i.e. model checking and theorem proving) is not mentioned of the paper. The authors write about FSMs, this is a close thing but not so powerful and I know some formal models that were derived from protocols definitions and the sequence of protocol steps were encoded in a CSP way and finally, checked against properties encoded in the form of logics (i.e. with the SPIN tool and Promela language usage). The results are proven on all possible situations and processes interleavings of the protocol model. The same can be said on modeling message passing using the Coq and Isabelle/HOL.
3. The "predictions of the future" that done in the end of the paper not so scientific and the paper lacks (in the reviewer's opinion) for the usage of formal logics for the definitions of protocols requirements as a next step from NLP-expressed requirements as well as testing blockchain protocols.
Also, the paper lacks for some real-world examples when fuzzing really helped to find a bug in a protocol that has been mentioned in CVE (common vulnerabilities database), with the explanations of the nature of the bug and how it has been found.
The title has two "of". The first one can be changed to "on" (but I do not sure).
"In 2009, Sulley was released on the GitHub platform [24]. Sulley is a fuzzing framework 210" (and some similar cases) -> It can be written in one phrase, that the Sulley fuzzing framework was released.
"Figure 3. Quantity change of network protocol fuzzing techniques." - Probably, it is better to reformulate.
The English in the paper is very simple and can be enriched.
Reviewer 2 Report
The paper surveys the development of fuzzing techniques and their issues. They use the generation of test cases and testing conditions to summarize the different fuzzing techniques.
However, the paper misses some important items;
- the paper must pose the research questions the survey aims to resolve. We need to know by the end of the paper if those questions are answered or not.
- As a survey paper, the manuscripts lack in-depth comparisons of existing approaches. For example, what are the advantage and disadvantages of different approaches?
- Being the paper a literature survey, it is necessary to provide all methodological details so that the reviewers can assess the quality. In concrete, it is necessary to include at least the search string, the digital libraries that were searched, a summary of papers found/filtered out, and the data analysis approach that was applied. In addition, an attachment or document available online with the papers considered at each stage (even the discarded ones) is a good practice for this type of paper.
Minor editing of English language required
Reviewer 3 Report
The authors of this survey paper addressed the gap of protocols fuzzing by categorizing and summarizing research in protocol fuzzing, the work is satisfactory, however a major overhaul is needed to improve its readability and contribution. Here are some notes that may be needed to be addressed: 1. Further analysis of the current survey work (ref [7,13,14]) need to be included thus the gaps on them are emphasized and the contribution of this work is highlighted 2. The authors didn’t highlighted nor explained the most relevant techniques in the state of the art. They should include some arguments and discussion relevant to the algorithms. More comparison and analysis of the work are needed. Thre is also lack of any pictorial illustration of any aspect of the survery 3. There are many summaries subsections, it is need but the authors need to include those summaries in the body of the sections and at the end they may introduce a discussion section to go through the summaries, the paper structure needs to be improved 4. The future directions need to be treated fairly and further context needed to be added with this regard may be in a separate section 5. Conclusions need to be improved 6. The survey needs to be supported with more references
satisfactory
Round 2
Reviewer 2 Report
Thank you for re-submitting a revised manuscript.
The authors addressed the different comments.
Minor editing of English language required
Reviewer 3 Report
Comments are addressed